# Investigation of the Effect of the Shape of Cutting Knives Limiting Burr in High-Strength Multiphase Steel Sheets

**DOI:** 10.3390/ma18020282

**Published:** 2025-01-10

**Authors:** Sebastian Mróz, Andrzej Stefanik, Piotr Szota, Sabina Galusińska, Dariusz Zaława, Andrzej Adamiec, Natalia Zaława

**Affiliations:** 1Faculty of Production Engineering and Materials Technology, Czestochowa University of Technology, 42-201 Częstochowa, Poland; andrzej.stefanik@pcz.pl (A.S.); piotr.szota@pcz.pl (P.S.); sabina.galusinska@pcz.pl (S.G.); 2Dar Stal Dariusz Zaława, ul. Przemysłowa 7, 42-300 Myszków, Poland; dzalawa@darstal.eu (D.Z.); aadamiec@darstal.eu (A.A.); nzalawa@darstal.eu (N.Z.)

**Keywords:** sheet metal, cutting knives, burr, numerical simulation, FEM

## Abstract

In this study, numerical modeling and experimental tests of the sheet metal cutting process were carried out in order to determine the shape of the cutting knives for a roller shear, ensuring the minimization of burr on the cut edge. A rolling mill was used for the tests, enabling the replication of the cutting process in a roller shear (demonstrating the possibility of using cutting rollers). The cutting edges of the sheets were examined using light microscopy and then compared with the results of numerical simulations to determine the cutting quality. The tests were performed for multiphase Complex Phase (CP) grade steel. The initial thicknesses of sheets were equal to 1 and 2 mm. Based on the results of theoretical research, four shapes of cutting rollers were designed, of which two shapes were selected for experimental tests. The analysis of the test results shows that the lowest burr values were obtained for straight and beveled rollers. Analyzing the size of burr obtained in experimental tests, it can be concluded that for each of the two variants of the roller shape, a reduction in burr was achieved. Greater reductions in burr were achieved for shaped (cut) rolls.

## 1. Introduction

Sheet metal is one of the most widely used input materials for the manufacture of many products, including structures or pipes and sections. Nowadays, high-strength multiphase steels are increasingly being used to manufacture the aforementioned products, mainly in the automotive industry [1,2]. An indisputable advantage of multiphase steels is their high strength, reaching up to 1500 MPa with relatively high ductility, thanks to which they are used in plastic processing, including stamping or bending [3,4]. Despite the numerous obvious advantages of the mechanical properties of AHSS, they have a significant disadvantage, which is difficulties with processing CP and DP steels, including cutting processes, which limits their widespread application. The appropriate selection of process parameters allows their repeatable and predictable plastic and mechanical processing also on automated processing lines.

One of the basic steps in the manufacture of sheet metal products is the cutting operation using roller shears. Cutting is a process that violates the cohesion of the material (metal), and the end result of cutting is the complete or partial separation of one part from another [5]. Different cutting methods are applied depending on the adopted sheet metal forming technology and its intended use. The most important ones are mechanical cutting [6], gas cutting [7], laser beam cutting [8], plasma cutting [9] and water jet cutting [10]. For thin sheets, cutting is usually performed with a guillotine or roller shear [11,12]. Using a roller shear is one of the cheapest and most efficient ways to cut sheet metal. The process of cutting sheet metal with a roller shear consists of several phases, namely, the elastic deformation phase, the elastic–plastic deformation phase, the plastic flow phase, the fracture phase and the phase of complete separation of the cut piece from the sheet metal. The extent to which the respective phases of cutting sheet metal with two cutting elements occur depends on the thickness of the material, the elastic–plastic properties of the sheet metal and the geometry of the cutting tools. The quality of the cut surfaces is very important in sheet metal cutting processes using roller shears. The increasing requirements of the steel market for quality and production efficiency force the use of cutting parameters that eliminate, or greatly reduce, the possibility of defects on the cut surface of the following types: burrs, edge bending and high roughness of the surface across the thickness and width of the sheet [13]. For example, in the production of cold-formed pipes and sections, the quality of the surface of the cut sheet significantly affects the speed and quality of welding of pipes and sections.

The mechanism and optimization of sheet metal cutting in terms of the surface quality of the cut sheet is a complex issue [14,15,16,17]. Thus, numerical methods are now being used to optimize the cutting process, enabling simulation of the sheet metal cutting process. These simulations can include actual material, physical and technological considerations (tool shape, cutting speed, selection of the appropriate value of knife spacing and friction coefficient). Reference [13] shows that these parameters have a very strong influence on sheet deformation, crack shape, tool wear and the quality of the resulting cut surface. Another quality parameter is the geometric accuracy of the sheet shape after cutting [18,19,20]. Reference [13] shows that the value of the cutting edge angle of the top knife α and the value of the clearance determine the height of the burrs on the cutting surface. It has been shown that increasing the angle α can cause an increase in burr by over 100% (for the range of angles from 1° to 6°), while increasing the distance between cutting edges from 0.01 mm to 0.2 mm results in an increase in burr by over 600%. The authors of [15] have shown that increasing the hardness of the sheet metal by heat treatment reduces the size of the burr from 4 mm thick 22MnB5 steel when cut on a guillotine; it decreased from 52.20 µm in the basic condition to 15.50 µm for the steel in the after-hardening condition. Studies presented in References [21,22,23] confirmed that the quality of the surface after cutting depends on a number of process parameters, such as the spacing between knives in the horizontal and vertical directions, lubrication method, shearing speed, knife geometry and the method of clamping the sheets. The effect of these parameters on the process is not fully clear, making it difficult to control the cutting process. Thus, in industrial practice, proper configuration of cutting parameters (combined with experience) is achieved by trial and error. As a result, the final product often has serious flaws and defects on the cut edge, such as burrs and wavy edges. The inadequate selection of the parameters of the cutting process can result in bending of the sheet metal, its deformation on the cross-section, the possibility of burr formation and an increase in the height of burrs [23]. An excessive burr value after the cutting process may result in the need to perform an additional technological operation of edge deburring. When deburring, all protruding remnants are removed from the edges of the workpiece. Sheets can be deburred either manually or mechanically [24,25]. The deburring process generates additional costs and reduces production efficiency. One of the cutting processes that allows for reducing the size of the burr is the cutting process on roller shears. Following these expectations, cutting tool suppliers are producing roller shears with increasingly better performance characteristics, but their shape in most cases is simple, i.e., the cutting edge in the roller ends with a flat surface at 90 degrees to the vertical edge [25]. Dienes [26] provides an example of a comprehensive approach to the design of a roller shear cutting process. It is the world leader in the design and manufacture of cutting tools, and in its opinion, the value of the gap between cutting knives (clearance), knife blade durability and burr generation are influenced by four important geometric relationships, namely, the cutting point, shear angle (roller inclination angle or alignment), knife overlap and dimensional runout. It is clear from the above that a valid approach is to select the shape of the knife part of the cutting rollers ensuring minimization of burrs on the cut edge. The shear angle creates a cutting point by angling the upper blade of the knife to the lower blade. One side of the material in the cutting slot must pass across and at an angle. The angle is determined by the knife holder and should be set to the minimum required for the material (the angle range is ½ to ¾ of a degree). Because the top knife is thin and very sharp, the contact at the shear angle through the point on the bottom edge of the knife is very small, causing the knife edge to wear quickly until a sufficient surface is formed that can support the lateral force load [26]. Many different options for beveling the top knife blade are currently in use. The factors determining the use of a particular type of blade include the type of metal being cut, the speed of action and side load and the force required.

Summarizing the analysis of the state of the art on the methods and parameters of cutting using roller shears, the main parameters determining the quality of cutting sheet metal include the type and thickness of the sheet metal, the size of the gap between the rollers and the angle of the rollers, the speed of cutting and the shape of the cutting knives. To optimize the cutting process, it seems necessary to use numerical methods to analyze the main cutting parameters. This reduces the time for designing the sheet metal cutting technology using roller shears and contributes to improving the quality of the cutting edge, including reducing the size of burrs. The aim of the work carried out by the authors is to develop a comprehensive technology for manufacturing CP steel pipes, in which one of the operations is cutting the sheet metal on roller shears (Figure 1). In industrial conditions, the sheet metal is simultaneously cut into several strips with dedicated widths corresponding to the given final diameter of the pipe. Thus, in this study, based on theoretical research using numerical modeling and experimental studies, the shape of cutting knives (roller shears) limiting the burrs on the edges of sheet metal after the cutting process was designed.

## 2. Materials and Methods

High-strength multiphase CP steel of grade CR570Y780T-CP according to VDA.239-100 [27] was used for the tests. The chemical composition and properties of the steel used in the tests are listed in Table 1 and Table 2.

As part of the study of the mechanical properties of the steel, static tensile tests were carried out to determine Young’s modulus using a ZWICK Z/100 testing machine (ZwickRoell, Ulm, Germany). In turn, essential tests to determine plastic melt curves were performed using the Gleeble 3800 metallurgical process simulator (Dynamic Systems Inc., Poughkeepsie, NY, USA). Plastometric tests were performed in tensile tests for a range of cold-forming temperatures (20; 100 and 200 °C) and 3 strain rate values (0.1; 1.0 and 10 1/s). Figure 2 shows the results of plastometric tests for the selected CP steel grade.

To describe changes in the value of *σ_f_* as a function of strain, temperature and strain rate, a number of functions are used, which can be found in publications [28,29], among others. In the present study, the Hensel–Spittel equation [30] was used to describe the flow stress function:(1)$$\sigma_{f}=K\cdot e^{m_{1}\cdot T}\cdot T^{m_{8}}\cdot \varepsilon^{m_{2}}\cdot e^{\frac{m_{4}}{\varepsilon }}\cdot {\dot{\varepsilon }}^{m_{3}}\cdot \varepsilon^{m_{6}T}\cdot {\left(1+\varepsilon \right)}^{m_{5}\cdot T}{\dot{\varepsilon }}^{m_{7}\cdot T}\cdot$$
where *σ_f_*—flow stress; *T*—temperature; ε—actual strain; $\dot{\varepsilon }$—strain rate, *K*; and *m*_1_÷*m*_8_—function coefficients.

Relationship (1) is often used for determining the value of *σ_f_* in computer programs for the numerical modeling of plastic processing. After approximating the plastometric results, the coefficients of Equation (1) were determined. The values of these coefficients are shown in Table 3.

The computer simulation of the cutting process was carried out with the use of a elastoplastic model in the triaxial state of strain by using the Forge^®^ NxT 2.1 program (Transvalor, Biot, France) based on the FEM (Finite Element Method), whereas the properties of the deformed material were described according to the Norton–Hoff [31,32] conservation law. This program enables the modeling of the cutting process using the limit values of the normalized Cockroft–Latham criterion [33] described by Equation (2):(2)$$C=C_{0}+\int_{0}^{\bar{\varepsilon }}\frac{\sigma_{1}}{\varepsilon_{i}}d\bar{\varepsilon }$$
where *C*_0_—initial value of the criterion, *ε_i_*—equivalent strain, *σ*_1_—maximum main tensile stress and *σ_i_*—equivalent stress

The criterion limit for which the loss of metal continuity is to occur is set by the user. Then, in each calculation step, the value of the analyzed criterion is determined and later compared with a previously set limit value. If the calculated value of the normalized Cockroft–Latham criterion for a given mesh element is greater than the critical value, the element is removed. When an element is removed, the connection table and element array are rebuilt.

The drawbacks of the element removal algorithm is the calculation error associated with loss of the deformed metal mass assigned to the removed elements. In order to minimize calculation errors, it is necessary to determine in advance the area in which the material may be lost. After that, the size of the mesh elements in this area must be modified. The most common solution used when modeling the fracture process is to use elements with the smallest side dimension in the immediate vicinity of the presumed fracture site.

In this study, a comparative method was used for determining the limits of the normalized Cockroft–Latham criterion. This method is commonly used for determining limiting values of the cracking criteria. The premise of the comparative method is to conduct experimental studies to determine characteristic numerical values for a given process (in the case of determining the value of the normalized Cockroft–Latham fracture criterion, it is elongation) and then to map the process assuming parameters corresponding to real conditions in numerical modeling. During modeling in the computer program, the values of the analyzed parameters are calculated with high computational accuracy. Tests were conducted for paddle-shaped specimens with varying width in the notch zone, 5 mm and 8 mm. The specimen shapes used make it possible to determine the effect of the strain path corresponding to the plastic forming process of the pipes for the studied sheets. The study was conducted using the Gleeble 3800 system (Dynamic Systems Inc., Poughkeepsie, NY, USA).

After stretching, the critical elongation at which the loss of continuity occurred for the tested sheet, propagating to complete rupture, was determined from the measured data obtained. The determined values of the normalized Cockroft–Latham criterion are shown in Table 4 (Δ—absolute error, δ—relative error).

Analyzing the data presented in Table 4, it can be noted that for both variants of the applied notched paddle-shaped specimen geometries, the obtained values of the analyzed fracture criterion are consistent (in the analyzed range of stress distribution, no significant change in the limit value for the investigated fracture criterion was recorded, and thus the influence of the stress state for the investigated range can be ignored). As for the effect of strain rate on the limits of the normalized Cockroft–Latham criterion for the studied CP steel, it is negligible (increasing the strain rate positively affects the range of plastic forming but to a very limited extent). For further calculations and verification of the technology, the values determined for the smallest strain rate, for which the value of the analyzed criterion is the lowest, were adopted. In this case, the assumed critical value will be met for the entire range of strain rates that can occur in the cut zone (at higher strain rates, cracking may occur later than at lower rates).

Static tensile tests were carried out to determine Young’s modulus values for the studied steel grade. Young’s modulus was calculated using the tangent method. The value of Young’s modulus for CP steel in the CR570Y780T-CP grade is 208 GPa.

Based on a literature review, it was assumed that the designed cutting rollers would have the same diameters and would be set at a distance of 0.1 for the sheet thickness of 1.0 mm and 0.2 mm for the sheet thickness of 2.0 mm. The rollers overlap by 2 mm. The selection of cutting rollers was initially based on the selection of four variants of shapes to ensure efficient cutting of metal sheets up to 2 mm thickness (the study investigated the process of cutting metal sheets 1 and 2 mm thickness). The selected four variants of cutting roller shapes were used for conducting computer simulations of the cutting process, in which the most optimal shape was determined. The optimization criterion was to achieve minimum burr on the edge of the sheet metal in the cutting area. The selection included rolls with (I) rectangular (classic straight), (II) trapezoidal, (III) left-slanted and (IV) right-slanted shapes. Figure 3 shows the shape and dimensions of the designed cutting roller shapes.

Inverse system dynamics was used for the numerical model to represent the actual cutting process. Moreover, the model was simplified to one pair of cutting rollers, while introducing boundary conditions corresponding to the real process (side supports replacing the actual tension of the steel being cut). A stationary volumetric model held by two supports was introduced, and the cutting rollers, in addition to rotary motion, were also put into linear motion corresponding to the movement of the sheet being cut in the actual process. The assembly used is shown in Figure 4. It shows the mathematical model of the sheet used for cutting (1), the shaped rollers set up according to the assumptions made for numerical testing (2) and the immovable supports holding the strand to prevent displacement of the cut sheet (3).

Cutting rollers with a diameter of 195 mm were used for the tests. The speed of the rollers was constant at 0.2 m/s, which corresponded to the operating conditions of the D150 laboratory rolling mill. The friction factor between the sheet being cut and the cutting roller was assumed to be a constant equal to 0.4. The thickness of the cut CP steel sheet was 1 and 2 mm.

Experimental verification of the process of cutting CP steel sheets was performed on a two-high D150 laboratory rolling mill (Figure 5), which was modified so that cutting knives selected by numerical simulations (two types of cutting knives) could be installed in place of rollers. The use of one set of cutting rollers causes difficulties in inserting and guiding the sheet metal (occurrence of torques). Therefore, in the case of laboratory tests, it would require the development of additional equipment. In order to simplify laboratory tests, it was decided to use two sets of cutting rollers arranged symmetrically, which limited the bending of the sheet metal in the transverse direction. The geometry of the rollers was changed to enable concentric mounting of cutting discs on the rollers. The rollers were equipped with flanges, to which pulleys were screwed together with the appropriate spacers. Adjustment of the spacing between the rings was carried out by thin plates inserted between the cutting discs and spacers. Horizontal adjustment of the upper and lower cutting discs relative to each other was carried out by precisely aligning the plain bearings in the horizontal axis.

## 3. Results and Discussion

### 3.1. Results of Theoretical Analysis of the Sheet Metal Cutting Process

Using the designed tool shapes, computer simulations were designed of the process of cutting sheet metal with rotary shears. Figure 6 shows the successive stages of the numerical modeling of the cutting process for one of the adopted variants. When analyzing the shape of the resulting cut shown in Figure 6, it can be seen that at the initial stage until the crack initiation, the edge of the sheet undergoes plastic deformation, so in order to evaluate the size of the burr formed on the lateral surface, it is necessary to take measurements for the established cutting process when the propagation and cutting of the sheet take place.

In order to determine the amount of burr formed on the side edges of the cut sheet, the finite element mesh shapes of the analyzed sheets after cutting, obtained in numerical studies, were exported to external files. The resulting mesh models were then loaded into CAD software (RhinoCeros v. 4.0) to properly position the sheet metal and then produce cross-sections. Figure 7a shows the loaded finite element mesh model of a cut sheet in CAD and the same model after rendering (the arrows indicate the locations used for cross-sections in the cut trace zone for the established cutting process). Figure 7b shows a displaced model of the sheet metal (position perpendicular to the cutting plane) with marked cross-section perpendicular to the surface of the sheet metal after cutting.

Analogously, cross-sections were made for the next obtained shapes of the cut sheet for all analyzed variants of roll shapes and both sheet thicknesses. Figure 8 and Figure 9 show the cross-sections obtained using the CAD software based on numerical testing.

Analyzing the obtained shapes of cross-sections of metal sheets after the cutting process, it can be noted that for all variants with trapezoidal rollers, the phenomenon of pulling the edge of the sheet after cutting into the gap between the rollers can be observed. This means that for trapezoidal rollers (Figure 3b), the longitudinal cut line is shifted deep into the gap between the rollers, which, for most roll spacing settings, will result in the formation of a fairly significant pull of metal from the cut sheet. Thus, it should be assumed that, for the conditions studied, the use of such rollers will cause deterioration in the quality of the edges of the sheets after cutting. In other cases, the size of the burr is smaller in the cases analyzed. For right-slanted rolls (Figure 8d and Figure 9d), a reduction in the size of the burr is observed; however, this is related to the simultaneous bending of the sheet edge, as expected.

In order to carry out a detailed analysis of the mechanism of burr formation on the edges of the cut sheet, distributions of strain intensity (Figure 10) and tangential stress σ_XY_ (Figure 11) were determined in the transverse plane for the same distances from the knife axis at which the sheet was not fully cut.

Analyzing the data shown in Figure 10, it can be seen that for the variants with straight rollers and left-sided inclined rollers, the strain intensity in the middle zone (at the cut line) is lower compared to the variant with a trapezoidal roller (marked with a red arrow, Figure 10b). Therefore, it can be assumed that for these variants, the strain penetrates less into the depths of the cut sheet, where the cutting process will occur with less strain in the gap between the cutting rollers. In the case of the variant with trapezoidal rollers in the cutting line, the strain penetrates deep into the sheared sheet causing edge pulling and thus increasing the size of the burr. For right-angled rolls, the highest intensity of deformation is observed at the point of contact between the rolls and the sheet to be cut (red arrow, Figure 10d), which then causes deformation and bending of the sheet to be cut before it is actually cut, as can be observed in Figure 8d and Figure 9d. The initial deformation of the sheet before cutting affects the displacement of the cut line of the sheet and thus increases the size of the burr.

Analyzing the distributions of Shear stress σ_XY_ shown in Figure 11, it can be seen that for a right-hand slant roll in its contact zone with the cut sheet, the orientation of the stress changes, which will affect the shift of the cut line in the direction in which the value of this stress will be close to zero. This can result in a change in the shape of the sheet after cutting because after the initial cut in the surface zone, the direction of propagation of the cut into the depth of the sheet will also shift. A similar distribution of stress σ_XY_ is observed for trapezoidal rollers, which can also affect the shifting of the cut propagation line deep into the sheet albeit to a lesser degree. On the other hand, the obtained distribution of σ_XY_ stress for straight rolls and left oblique rolls shows that for these cases the cut line will be close to vertical and will run in the zone between the cutting rolls.

In order to determine in detail the size of the burr on the edges of the cut sheet, three cross-sections were made for each of the analyzed variants, and then the obtained cross-sections were loaded into CAD software, and the analyzed dimensions were measured. Average values were then calculated. A list for 1 mm and 2 mm thick sheets is shown in Table 5 (where L1–L3, P1–P3—measurement number on the sheet length, L—left part of the sheet, right part of the sheet). Analysis of the obtained results confirmed that it is not advisable to use trapezoidal-shaped rollers (Figure 3b), causing the edges of the sheet to be pulled. For the other variants, a reduction in the size of the burr on the edges of the cut sheet was achieved for both 1 mm and 2 mm thick sheets. In addition, for trapezoidal rollers, where a large amount of sheet edge burring was observed for one edge, while only edges with more burrs were considered for the averaged results. Based on the numerical results obtained, straight-shaped rollers (Figure 8a and Figure 9a) and left-slanted rollers (Figure 8c and Figure 9c) were selected for further experimental studies.

### 3.2. Experimental Analysis of the Sheet Metal Cutting Process

In the experimental tests, three specimens of 50 × (thickness of 1 or 2 mm) × 500 mm length each were cut longitudinally in the rolling direction from metal sheet coils. After cutting, three smaller samples were cut out of the samples from the beginning, middle and end parts (it was assumed that the zone in which the cutting process may be unstable does not exceed 50 mm at the beginning and end of the sheet being cut), and then metallographic microsections were taken from them, and the size of the burr was measured using an optical microscope. Based on measurements of the three samples taken, the average burr value was determined for the respective variants of cutting and thicknesses, from which sheets were made. Figure 12 and Figure 13 show selected views of the microsections for which burr size measurements were taken. The results of burr size measurements are summarized in Table 6 and Table 7 (where L1–L3, P1–P3—measurement number on the sheet length, L—left part of the sheet, right part of the sheet for the selected places).

From the data shown in Figure 12 and Figure 13, as well as in Table 6 and Table 7, it can be concluded that, regardless of the thickness of the cut sheet, the smallest burr values were obtained for rolls made according to the left-slanted beveled variant. The difference in favor of left-slanted beveled rolls was about 2.5% for 1 mm thick sheets and 7% for 2 mm thick sheets compared to the values obtained for straight rolls. Based on the analysis of experimental and numerical results, there is a high convergence of the obtained results. In none of the analyzed cases did the difference exceed 10%, which indicates the correct choice of initial conditions for numerical modeling. In order to reduce the error value and obtain a more suitable shape of the burrs after cutting, it may be necessary to introduce modifications to the theoretical model. However, due to the practical application of the research results presented in this paper, this stage should be carried out after industrial verification. The benefit of reducing the burr size is the ability to apply less pressure to the edge of the strip when welding the slotted pipe. This should lead to lower energy consumption and reducing the number of products that do not meet acceptance standards.

## 4. Conclusions

The aim of the study was to design the shape of cutting rolls for slitting sheets made of CP steel to ensure minimal burrs. The theoretical research included numerical modeling, based on which four shapes of cutting rolls were developed, of which two shapes were selected for experimental testing. An analysis of the test results shows that the smallest burr values were obtained for the straight and left beveled rolls. Analyzing the magnitudes of the burr obtained in the experimental studies, it can be concluded that for each of the two variants of the shape of the rolls, the burr size was below 0.06 mm for 1 mm sheet metal and 0.12 mm (limit values according to standards). Greater burr reduction was achieved for shaped rolls (compared to straight rolls, this is an almost 20% reduction in the burr size for 1 mm thick sheets and about a 5% reduction for 2 mm thick sheets), and this shape of roll was recommended for implementation in industrial settings. Reducing burr in cut sheets of steel for pipe production significantly affects the quality of induction welding of the finished pipe, which is related to the energy demand. This results in a significant reduction in the number of finished products that do not meet acceptance standards.

## Figures and Tables

**Figure 1 materials-18-00282-f001:**
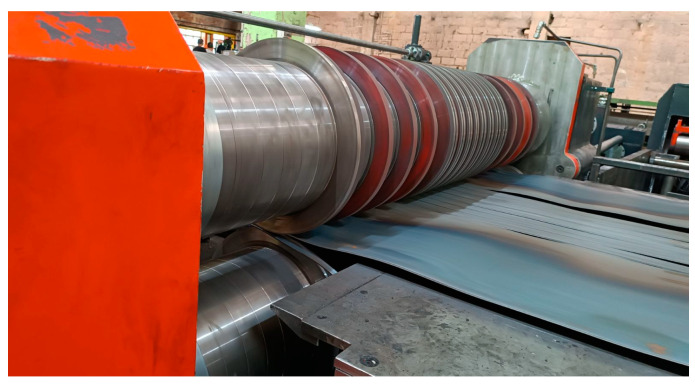
An example of a longitudinal cutting of sheet metal from a coil.

**Figure 2 materials-18-00282-f002:**
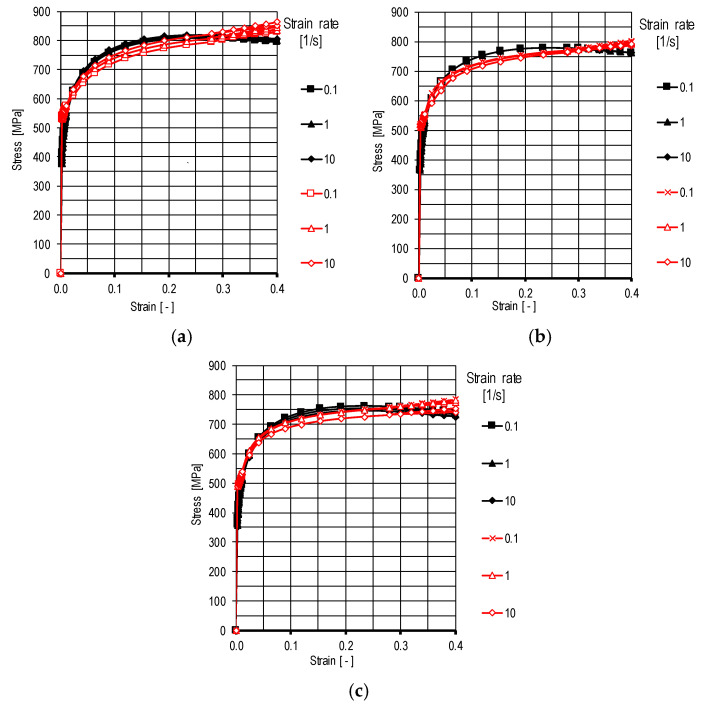
Plastic flow curves for CP steel in CR570Y780T-CP grade: (**a**) temperature of 20 °C, (**b**) temperature of 100 °C, (**c**) temperature of 200 °C, red color—results of plastometric tests, black color—results of approximation of plastometric tests.

**Figure 3 materials-18-00282-f003:**
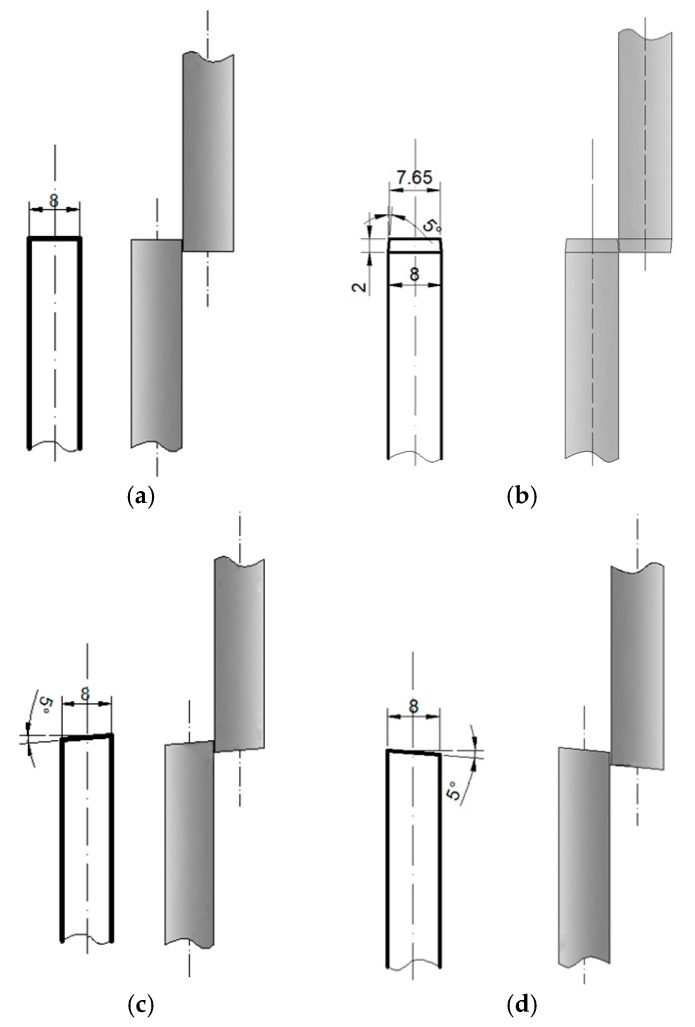
Dimensions and shape of cutting rollers: (**a**) rectangular (classic straight), (**b**) trapezoidal, (**c**) left-slanted, (**d**) right-slanted.

**Figure 4 materials-18-00282-f004:**
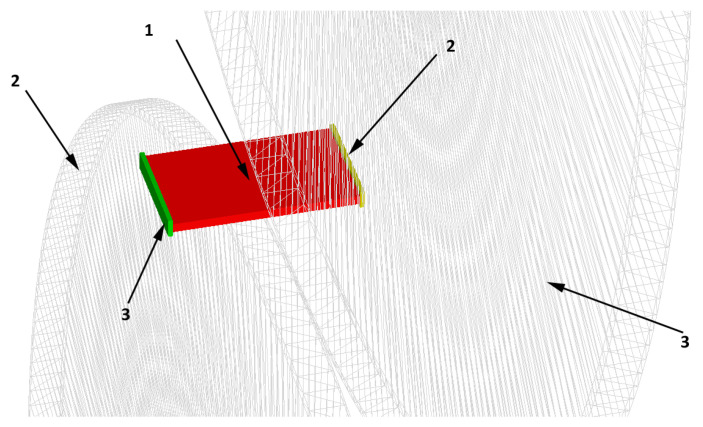
Assembly of numerical model elements used for sheet metal cutting analysis.

**Figure 5 materials-18-00282-f005:**
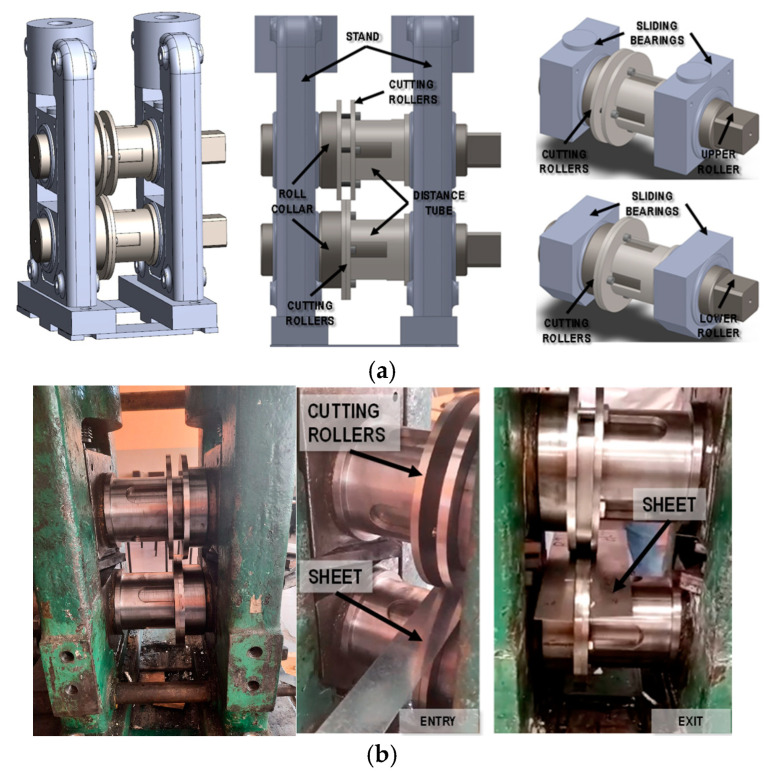
View of the (**a**) model of the two-high D150 laboratory rolling mill, (**b**) the rolling mill modified to work as a circular shear for cutting sheet.

**Figure 6 materials-18-00282-f006:**
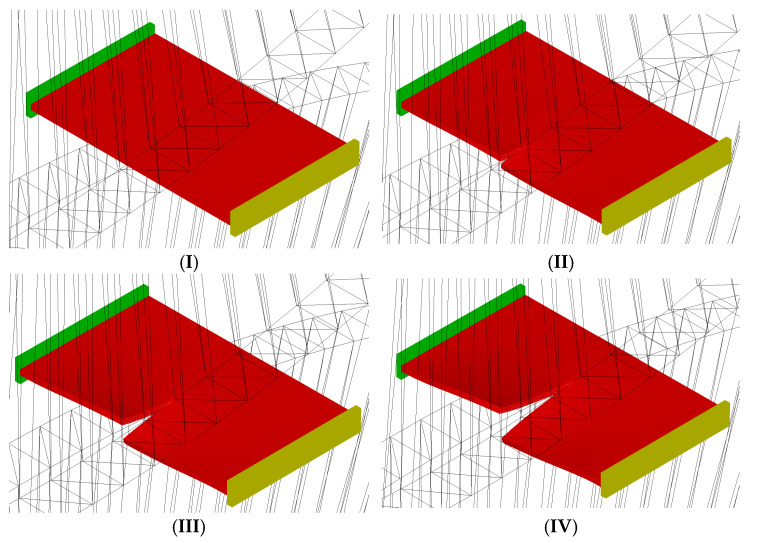
The subsequent stages (**I**–**IV**) of sheet metal cutting using shaped rollers for an example of a sheet metal cutting variant.

**Figure 7 materials-18-00282-f007:**
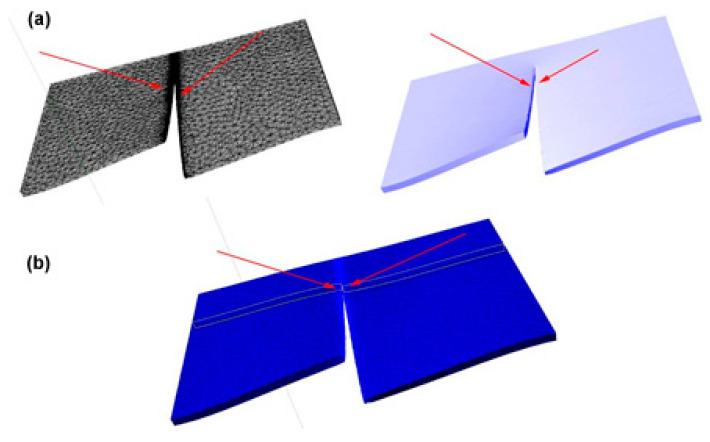
Finite element mesh model of the sheet metal after cutting—(**a**), mesh model of the cut sheet metal with a cross-section made for measuring the burr size—(**b**).

**Figure 8 materials-18-00282-f008:**
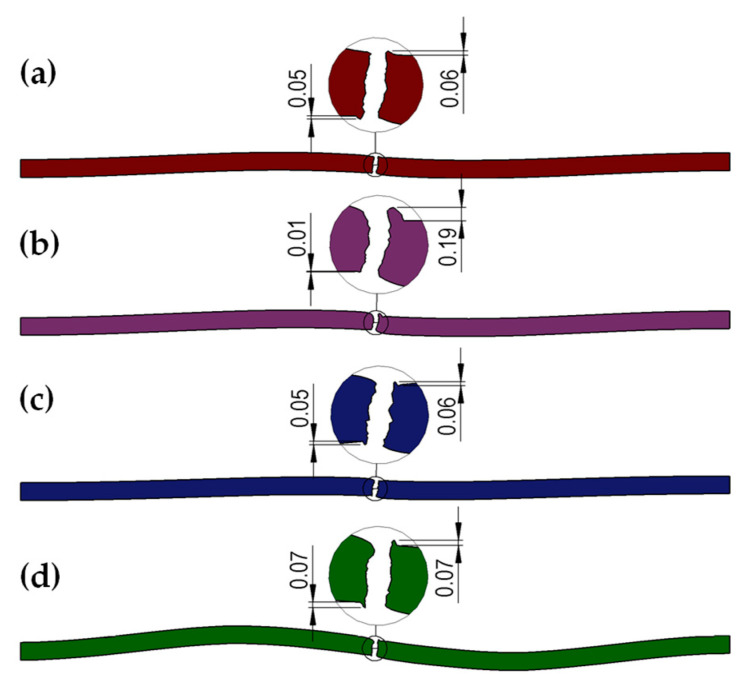
Shapes and dimensions of cross-sections of CP steel sheet of grade CR570Y780T-CP with a thickness of 1 mm obtained in theoretical tests after cutting for rolls: (**a**) rectangular (classic straight), (**b**) trapezoidal, (**c**) left-slanted, (**d**) right-slanted.

**Figure 9 materials-18-00282-f009:**
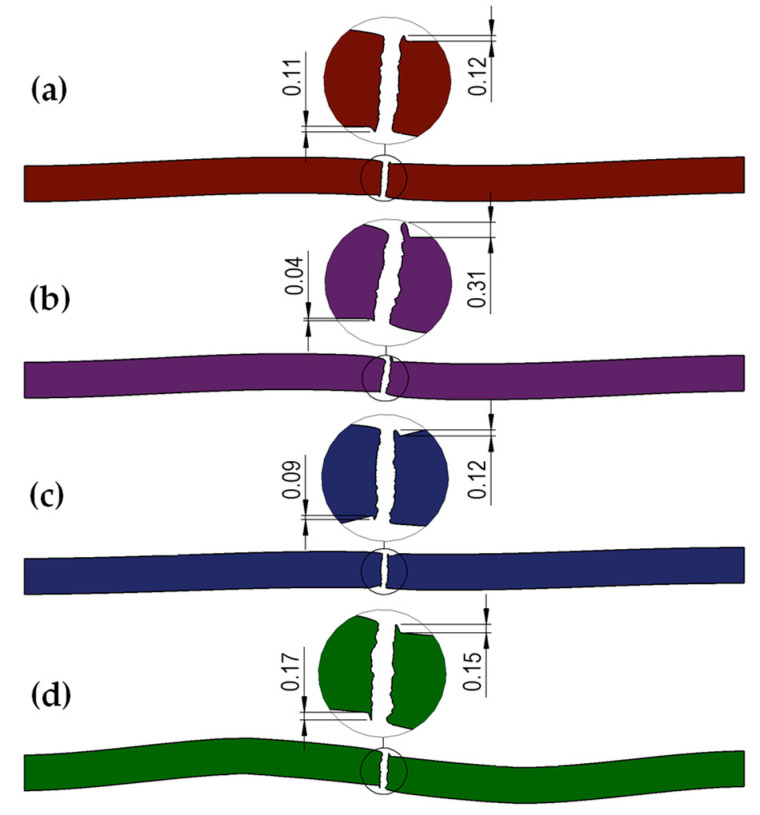
Shapes and dimensions of cross-sections of CP steel sheet of grade CR570Y780T-CP with a thickness of 2 mm obtained in theoretical tests after cutting for rolls: (**a**) rectangular (classic straight), (**b**) trapezoidal, (**c**) left-slanted, (**d**) right-slanted.

**Figure 10 materials-18-00282-f010:**
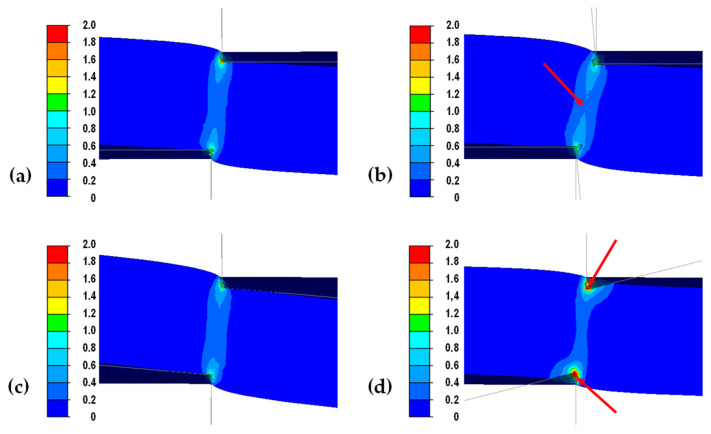
Distribution of effective strain on the cross-sections of 2 mm thick sheet metal obtained in theoretical tests: (**a**) straight rollers, (**b**) trapezoidal rollers, (**c**) left-slanted rollers, (**d**) right-slanted rollers.

**Figure 11 materials-18-00282-f011:**
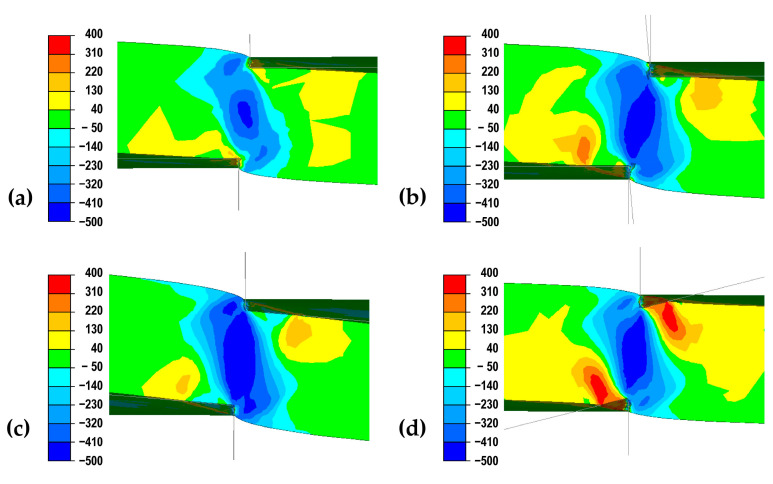
Shear stress *σ_XY_* distributions on the cross-sections of 2 mm thick sheet metal obtained in theoretical tests: (**a**) straight rollers, (**b**) trapezoidal rollers, (**c**) left-slanted rollers, (**d**) right-slanted rollers.

**Figure 12 materials-18-00282-f012:**
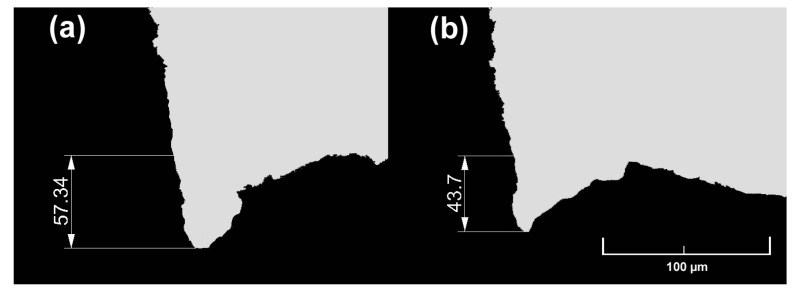
Selected shapes and dimensions of the burr obtained in experimental tests for sheets of 1 mm thickness: (**a**) straight roller, (**b**) left-slanted roller (dimensions in µm).

**Figure 13 materials-18-00282-f013:**
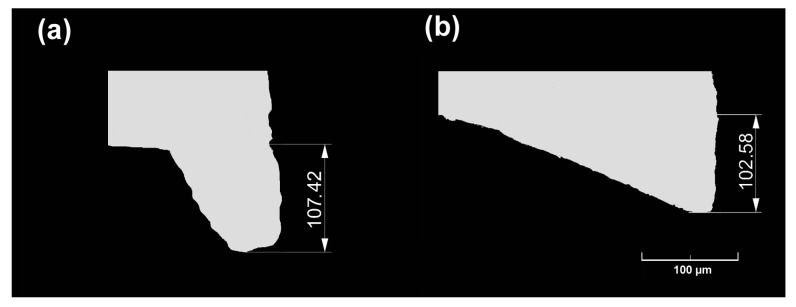
Selected shapes and dimensions of the burr obtained in experimental tests for sheets of 2 mm thickness: (**a**) straight roller, (**b**) left-slanted roller (dimensions in µm).

**Table 1 materials-18-00282-t001:** Chemical composition of the steel used for the tests.

Chemical Composition, % Mass.							
C	Si	Mn	P	S	Al	Cr + Mo	Ti + Nb
0.18	1.0	2.5	0.05	0.01	0.015–1.0	1.0	0.15

**Table 2 materials-18-00282-t002:** Mechanical properties of the steel used for tests.

Ys_0.2_, MPa	UTS, MPa	YM, GPa	E, %
570–720	780–980	208	10

**Table 3 materials-18-00282-t003:** The values of the *K* and *m*_1_÷*m*_8_ parameters used to determine the *σ_f_* value.

*K*	*m* _1_	*m* _2_	*m* _3_	*m* _4_	*m* _5_	*m* _6_	*m* _7_	*m* _8_
0.684788	−0.00721633	0.342418	0.02864	−0.00439388	−0.08198	0.000230534	0.0002181	1.41094

**Table 4 materials-18-00282-t004:** Limit values of the normalized Cockroft–Latham criterion determined in the tensile test for CP steel grade CR570Y780T-CP.

Strain Rate [s^−1^]	Limit Values of the Normalized Cockroft–Latham Criterion		Absolute Error Δ	Relative Error δ
	Notch 5 mm	Notch 8 mm		
0.1	0.518	0.521	0.009	1.74%
1.0	0.525	0.531		
10.0	0.539	0.537		

**Table 5 materials-18-00282-t005:** Burr values determined in theoretical tests for sheets with a thickness of 1.0 and 2.0 mm.

Roller Type	Measur. No. 1 [mm]		Measur. No. 2 [mm]		Measur. No. 3 [mm]		Average
	L1	P1	L2	P2	L3	P3	
1.0 mm thick sheet metal							
straight	0.05	0.05	0.05	0.06	0.04	0.05	**0.05**
trapezoidal	0.02	0.15	0.01	0.19	0.02	0.18	0.17
left-slanted	0.05	0.07	0.05	0.06	0.05	0.06	**0.06**
right-slanted	0.08	0.07	0.07	0.08	0.07	0.06	0.07
2.0 mm thick sheet metal							
straight	0.10	0.11	0.12	0.11	0.12	0.12	0.11
trapezoidal	0.04	0.35	0.04	0.31	0.07	0.35	0.33
left-slanted	0.11	0.12	0.09	0.12	0.12	0.11	0.11
right-slanted	0.14	0.15	0.17	0.15	0.15	0.16	0.15

**Table 6 materials-18-00282-t006:** Burr values determined in theoretical and experimental tests for sheets with a thickness of 1.0 mm.

Roller Type	Measurement Results, [mm]						av. Lab. Test [mm]	av. FEM [mm]	Difference Lab./FEM [%]
	L1	P1	L2	P2	L3	P3			
straight	0.0487	0.0421	0.0456	0.0528	0.0485	0.0501	0.0476	0.053	9.5
	0.0459	0.0471	0.0446	0.0503	0.0478	0.0511			
	0.0456	0.0457	0.0513	0.0465	0.0477	0.0459			
left-slanted	0.0487	0.0492	0.0512	0.0437	0.0421	0.0474	0.0471	0.051	7.7
	0.0507	0.0511	0.0476	0.0421	0.0438	0.0436			
	0.0483	0.0502	0.0469	0.0485	0.0468	0.0454			

**Table 7 materials-18-00282-t007:** Burr values determined in theoretical and experimental tests for sheets with a thickness of 2.0 mm.

Roller Type	Measurement Results, [mm]						av. Lab. Test [mm]	av. FEM [mm]	Difference Lab./FEM[%]
	L1	P1	L2	P2	L3	P3			
straight	0.1083	0.1091	0.1138	0.1123	0.1098	0.1103	0.1106	0.121	8.6
	0.1105	0.1074	0.1056	0.1082	0.1104	0.1083			
	0.1142	0.1123	0.1094	0.1107	0.1172	0.1123			
left-slanted	0.1053	0.1014	0.1064	0.1026	0.1012	0.1035	0.1034	0.110	6.0
	0.1026	0.1094	0.1023	0.1043	0.1022	0.1051			
	0.1003	0.1014	0.1021	0.1009	0.1043	0.1052			

## Data Availability

The original contributions presented in this study are included in the article. Further inquiries can be directed to the corresponding authors.

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
