# Peer review of "Investigation of the Effect of the Shape of Cutting Knives Limiting Burr in High-Strength Multiphase Steel Sheets"

_materials, 2025, doi:10.3390/ma18020282_

Round 1

Reviewer 1 Report

Comments and Suggestions for Authors

The presented article dealing with the influence of the shape of cutting blades on sheet metal cutting begins by analyzing a sufficient amount of professional literature from various publications. The following chapters follow each other methodologically and thematically and present a brief professional review and the results of experimental measurements. Selected parameters are also compared with model designs. The article is of a good level in terms of graphics with appropriately selected images and created graphs.

Comments on the article:

- At the end of the paragraph in line 247, the Young's modulus is given with a value of 183200 MPa. Is this value correct?

- Figure 2 presents models and variants of cutting blades where there are two cutting cylinders, while the photographs of real shears (Figure 4) already have three cylinders. It is necessary to describe why the model had two and the real one had three cutting cylinders.

- The description of Figures 7 and 8 is the same, which is probably not true. It needs to be corrected or explained.

- The numbering of the tables is not consecutive (table 5 is marked twice), and at the same time, a square bracket is incorrectly inserted in tables 5 and 6. Formatting correction is required.

- There is a lack of a more detailed methodology of the experiments performed, from which it would be clear whether the results are from repeated or random measurements. Although it is stated that the measurements were made at the beginning, in the middle and at the end of the experimental samples, it is necessary to describe the reliability and accuracy of the measured values.

- What effect does the width of the cutting discs have on the deformation of the cut material?

- The measurement results are only briefly described, which would need to be described in the discussion and a more detailed conclusion.

Based on my assessment of the submitted analysis of the article, I recommend that the author make minor changes.

Author Response

  1. At the end of the paragraph in line 247, the Young's modulus is given with a value of 183200 MPa. Is this value correct?

The Young's modulus value shown was incorrect for this steel grade. The correct value is 208 GPa, value was corrected. Thanks for pointing it out. The correct value was used for the research, while the value obtained for other experimental tests performed on the Zwick Z100 testing machine was included in the editing.

  1. Figure 2 presents models and variants of cutting blades where there are two cutting cylinders, while the photographs of real shears (Figure 4) already have three cylinders. It is necessary to describe why the model had two and the real one had three cutting cylinders.

In the case of numerical modelling, a sheet metal section with two cutting rollers was used, which corresponds to the actual cutting conditions in the industrial process. In the industrial process, the unrolled sheet metal from the coil is cut into several adjacent strips, which are then rolled into independent coils of smaller width. In the industrial process, due to the tension of the cut strip and due to the fact that the sheet metal is cut at the same time into several finished widths, there is no problem of bending individual sheet metal strips.

In the case of the numerical model, due to the need to simplify the model due to the limitation of the number of finite elements and the calculation time, additional side tools were introduced to hold the cut sheet metal in a limited way in a constant horizontal transverse position.

In the case of laboratory verification, this effect was achieved by introducing a double set of cutting knives.

Explanations have been added to the article text:

“The inverse system dynamics was used for the numerical model to represent the actual cutting process. Moreover, the model was simplified to one pair of cutting rollers, while introducing boundary conditions corresponding to the real process (side supports replacing the actual tension of the steel being cut). A stationary volumetric model held by two supports was introduced, and the cutting rollers, in addition to rotary motion, were also put into linear motion corresponding to the movement of the sheet being cut in the actual process.”

“The use of one set of cutting rollers causes difficulties in inserting and guiding the sheet metal (occurrence of torques). Therefore, in the case of laboratory tests, it would require the development of additional equipment. In order to simplify laboratory tests, it was decided to use two sets of cutting rollers arranged symmetrically, which limited the bending of the sheet metal in the transverse direction. The geometry of the rollers was changed to enable con-centric mounting of cutting discs on the rollers.”

  1. The description of Figures 7 and 8 is the same, which is probably not true. It needs to be corrected or explained

Changes have been made to the text.

  1. The numbering of the tables is not consecutive (table 5 is marked twice), and at the same time, a square bracket is incorrectly inserted in tables 5 and 6. Formatting correction is required.

Changes have been made to the text.

  1. There is a lack of a more detailed methodology of the experiments performed, from which it would be clear whether the results are from repeated or random measurements. Although it is stated that the measurements were made at the beginning, in the middle and at the end of the experimental samples, it is necessary to describe the reliability and accuracy of the measured values.

The tests were carried out for a selected type of sheet metal for six samples that were cut longitudinally. Every second cut strip was selected for the tests, from which samples were taken from the middle part of the coil (it was assumed that at least 50 mm should be measured at both the beginning and the end of the cut sample to avoid a zone in which the cutting process may be unstable, this corresponds to the actual conditions of the cutting process where the beginnings and ends are cut off after cutting). The tables present average results for the middle areas of the tested samples, along with the measurement deviations occurring for different samples. The samples were measured at 3 points, based on which the average was calculated.

Changes have been made to the text.

  1. What effect does the width of the cutting discs have on the deformation of the cut material?

The research did not analyze the influence of the roller width on the deformation of the cut material, standard widths of commercially available rollers used by Darstal for cutting sheets of other types of steel were assumed.

However, it can be assumed that with the increase of the roller width, the zone of interaction of the tools with the cut material will increase, which in the case of shaped rollers may cause increased bending of the edges, causing uncontrolled deformation of the cut sheet.

Precise determination of the influence of the roller width on the deformation of the cut sheets would require performing at least several variants of additional theoretical tests for different shapes and widths of the cutting rollers.

  1. The measurement results are only briefly described, which would need to be described in the discussion and a more detailed conclusion.

Changes have been made to the text.

Reviewer 2 Report

Comments and Suggestions for Authors

I would like to recommend this manuscript for publication after minor revision:

1. For a research article, the Introduction is too long;

2. The first keyword should be revised as "sheet metal"?

3. The full name of FEM should be provided;

4. The curve in Figure 5 is too thin and the resolution is not high, making it difficult to see clearly;

5. Please add scale bar to Figure 11 and Figure 12.

Author Response

  1. For a research article, the Introduction is too long.

Due to other reviews, the introduction was revised.

  1. The first keyword should be revised as "sheet metal"?

Changes have been made to the text.

  1. The full name of FEM should be provided.

Changes have been made to the text.

  1. The curve in Figure 5 is too thin and the resolution is not high, making it difficult to see clearly.

The figures in the text have been changed, the readability of Fig. 6 (previously Fig. 5) has been improved.

  1. Please add scale bar to Figure 11 and Figure 12.

Scale has been added to the corresponding images, currently Fig. 12 and Fig. 13

Reviewer 3 Report

Comments and Suggestions for Authors

This article provides information about an interesting issue. In my opinion, it is suitable for publishing after the correction according to the comments mentioned below:

 1) The literature review in the Introduction section provides a sufficient overview of qualitative insights. However, it should be supplemented with a quantitative perspective.

2) Similarly, the literature review would benefit from a greater emphasis on alternative solutions or innovations in cutting technologies, rather than focusing solely on roller shears.

3) In the Results section, it would be useful to include information about the environmental impact of the materials used and the cutting process.

4) Additionally, there is a lack of a detailed economic assessment, specifically regarding the impact of implementing the proposed solutions.

5) In the Discussion section, it is essential to explore the possibilities for automating the proposed solutions.

6) The Conclusion section should highlight the practical benefits and specific implications of the results. Since the study was conducted on a small scale, it is important to address how these findings may translate to broader industrial applications.

Author Response

  1. The literature review in the Introduction section provides a sufficient overview of qualitative insights. However, it should be supplemented with a quantitative perspective.

Due to other reviews, the introduction was revised. In many articles the issue of burr measurement is only mentioned, there are no complex studies on the changes in burr size depending on the cutting process parameters for the tested steels. Examples of measuring burr after cutting for various conditions are provided.

  1. Similarly, the literature review would benefit from a greater emphasis on alternative solutions or innovations in cutting technologies, rather than focusing solely on roller shears.

Due to other reviews, the introduction was revised. The literature review mentions other methods of cutting sheets, e.g. using plasma, laser or water. However, longitudinal cutting is the most efficient method for cutting coil sheets and therefore the study focused on this method. Due to the scope of the research, which was part of a project aimed at implementing a specific technological solution for the production of longitudinally welded pipes made of high-strength AHSS steel, only the study of the cutting process on roller shears as it is carried out at Darstal was limited. A commentary was introduced to the text of the article specifying the purpose, motivation and technological problem that guided the authors.

  1. In the Results section, it would be useful to include information about the environmental impact of the materials used and the cutting process.

As mentioned earlier, the presented research is a fragment of a larger technological problem, the final effect of which is the implementation of the technology of forming AHSS steel welded pipes in the Darstal company conditions. The presented research results confirm that the use of an effective method of longitudinal cutting of sheets after unwinding from the coil for the appropriate shape of the cutting rollers significantly reduces the burr generated in the process of cutting sheets into feed tapes for the process of shaping seam pipes, which facilitates the process of producing finished products in subsequent stages causing a significant reduction in the number of finished products that do not meet acceptance standards. The cutting process used is an efficient process, in which the use of appropriate roller shapes and working tool settings ensures its stability. The size of the burr significantly affects the process and quality of induction welding, which is related to energy demand. However, the energy effect was not recorded in the research.

  1. Additionally, there is a lack of a detailed economic assessment, specifically regarding the impact of implementing the proposed solutions.

The proposed solution is low-cost because it only requires the use of a different shape of cutting rollers for the cutting process (shaped rollers). The new shape of the rollers also increased their durability, which allowed them to be used longer without the need for regeneration. The benefit of reducing the size of the burr is the possibility of using less pressure on the edge of the tape during welding of the slotted pipe. This resulted in lower energy consumption.

  1. In the Discussion section, it is essential to explore the possibilities for automating the proposed solutions.

The study only analyzed the influence of the shape of the cutting rollers on the size of the burr produced. The industrial process itself is automated and, with appropriate parameters (roller shape, setting and kinetic parameters), it is stable and largely automated. Process automation includes, among others: on the selection of the tension of the cut sheet depending on the coil width, sheet thickness and steel type.

  1. The Conclusion section should highlight the practical benefits and specific implications of the results. Since the study was conducted on a small scale, it is important to address how these findings may translate to broader industrial applications.

In the case of the conducted research, as previously emphasized, they are an element of the implemented technology for manufacturing HS steel pipes. Specific shapes of the cutting rollers were used in the process of cutting sheet metal unrolled from a coil to the appropriate widths in industrial conditions. The presented results of small-scale research concerned only the shape of the cutting knives and are used in industrial technology. The new shape of the rollers also increased their durability, which allowed for their longer use without the need for regeneration. The benefit of reducing the burr size is the ability to apply less pressure to the edge of the strip when welding the slotted pipe. This led to lower energy consumption.

Round 2

Reviewer 3 Report

Comments and Suggestions for Authors

Dear Authors,

thank you very much for providing the revised version of your article. After your explanation, it is possible to publish this article in present form.